# Sleep-Related Declarative Memory Consolidation in Children and Adolescents with Developmental Dyslexia

**DOI:** 10.3390/brainsci11010073

**Published:** 2021-01-08

**Authors:** Flaminia Reda, Maurizio Gorgoni, Aurora D’Atri, Serena Scarpelli, Matteo Carpi, Erica Di Cola, Deny Menghini, Stefano Vicari, Giacomo Stella, Luigi De Gennaro

**Affiliations:** 1Department of Psychology, Sapienza University of Rome, 00185 Rome, Italy; flaminia.reda@uniroma1.it (F.R.); maurizio.gorgoni@uniroma1.it (M.G.); matteo.carpi@uniroma1.it (M.C.); erica.dicola@libero.it (E.D.C.); 2Department of Biotechnological and Applied Clinical Sciences, University of L’Aquila, 67100 L’Aquila, Italy; aurora.datri@univaq.it; 3IRCCS Fondazione Santa Lucia, 00179 Rome, Italy; serena.scarpelli@uniroma1.it; 4Child and Adolescent Psychiatry Unit, IRCCS Bambino Gesù Children’s Hospital, 00165 Rome, Italy; deny.menghini@opbg.net (D.M.); stefano.vicari@opbg.net (S.V.); 5Department of Life Science and Public Health, Catholic University of the Sacred Heart, 00153 Rome, Italy; 6Department of Education and Human Sciences, University of Modena and Reggio Emilia, 42121 Reggio Emilia, Italy; giacomo.stella@sosdislessia.com

**Keywords:** developmental dyslexia, neurodevelopmental disorders, Non-Rapid Eye Movement (NREM) sleep, memory consolidation, sleep spindles, sleep oscillations, learning disabilities, predictive neurocognitive factors, Slow Wave Activity, EEG topography

## Abstract

Sleep has a crucial role in memory processes, and maturational changes in sleep electrophysiology are involved in cognitive development. Albeit both sleep and memory alterations have been observed in Developmental Dyslexia (DD), their relation in this population has been scarcely investigated, particularly concerning topographical aspects. The study aimed to compare sleep topography and associated sleep-related declarative memory consolidation in participants with DD and normal readers (NR). Eleven participants with DD and 18 NR (9–14 years old) underwent a whole-night polysomnography. They were administered a word pair task before and after sleep to assess for declarative memory consolidation. Memory performance and sleep features (macro and microstructural) were compared between the groups, and the intercorrelations between consolidation rate and sleep measures were assessed. DD showed a deeper worsening in memory after sleep compared to NR and reduced slow spindles in occipito-parietal and left fronto-central areas. Our results suggest specific alterations in local sleep EEG (i.e., sleep spindles) and in sleep-dependent memory consolidation processes in DD. We highlight the importance of a topographical approach, which might shed light on potential alteration in regional cortical oscillation dynamics in DD. The latter might represent a target for therapeutic interventions aimed at enhancing cognitive functioning in DD.

## 1. Introduction

Human sleep plays a crucial role in learning and memory consolidation, through reorganization processes based on neural plasticity [1,2,3]. Specifically, for what concerns declarative memory, the beneficial effect of sleep in the integration of new information has been widely demonstrated in adults [3,4,5,6,7]. Two hallmarks of Non-rapid Eye Movement (NREM) sleep EEG have been recognized to be mainly involved in the consolidation of declarative memory: Slow Wave Activity (SWA) and sleep spindles [3,8,9].

A growing body of evidence suggests that sleep supports cognitive functioning and brain maturation during typical human development, a period characterized by intense plastic changes [9]. The relation between sleep and declarative memory, as well as the role of SWA and sleep spindles, has also been observed in children and adolescents, even if data are more conflicting [9]. Sleep problems also are an increased risk in several neurodevelopmental disorders [10,11,12]. Furthermore, the presence of sleep EEG alterations has been hypothesized to be related to the specific nature of the cognitive and behavioral phenotypes characterizing the different Neurodevelopmental Disorders (NDDs) (for a review see [10,13]). Besides, these alterations have been shown also to have an impact on memory functioning [14,15,16,17,18,19].

According to this general background, it is surprising that only one study [20] has investigated the relation between sleep features and memory consolidation in the most common neurodevelopmental disorder, which is Developmental Dyslexia (DD). This disorder is characterized by unexpected lower reading abilities for the chronological age, in presence of normal IQ and education [21] and its prevalence rates range from 5% to 15% [22]. The most established explanatory hypothesis about the etiology of DD suggests the presence of a phonological deficit as core feature of the disturbance [23]. However, in the last years, an interest towards a more complex definition of the disorder has been growing and recent models support a multiple deficit hypothesis [23,24,25].

Among the different cognitive domains involved, long-term memory seems to play a role in the disorder, even if the findings in this field are exiguous and controversial [26,27]. Concerning declarative memory, early findings showed alternatively the presence [28,29] or alternatively the absence [30] of a visual sequential memory deficit in populations with DD. Similar inconsistencies have been found in the verbal domain, with preserved [31] or impaired [30] long-term verbal abilities in children with DD. In university students with DD, cross-modal long-term memory problems have been reported in binding associations between pairs of phonological and visual features [32]. Similarly, in adults with DD, deficits in long-term memory representations have been documented [33], together with self-reported problems in remembering both facts [34] and episodically experienced events [35]. 

The assessment of sleep-dependent memory processes may help to better understand learning mechanisms in this population.

Sleep studies in DD are, to date, inconsistent and heterogeneous even if there is a general agreement on the presence of some kinds of alteration. Only one study [36] investigated sleep problems in DD, highlighting an increased incidence in this disturbance compared to normal readers (NR). Regarding sleep architecture, some studies found alterations in children with DD compared to NR [37,38,39], in terms of a decreased REM sleep and stage 3 NREM sleep percentage and an increase of stage 2 in children with DD compared to NR [37,38]. However, the most recent work in this field does not confirm these findings [20]. Moreover, microstructural abnormalities in DD, in particular an increase of sleep spindles and slow-wave activity (SWA), have also been observed [37,38]. 

Only one study, examining the potential impact of altered sleep on memory processes in DD [20], suggested a reduced role of sleep in declarative memory consolidation processes. While memory performance correlated with the expression of SWA (<1 Hz) in NR, no correlation was found for the group with DD. Moreover, children with DD showed a significant worsening after one week from the learning session, compared to the reading-abilities matched group, composed of NR younger children.

At present, given the well-established knowledge about the local nature of sleep [40,41], the main limitation of the literature about sleep in DD is the absence of data concerning the topographical distribution of its features. The aforementioned studies [20,37,38] did not analyze how the microstructural patterns are distributed on different cortical regions.

Starting from this background, the present study has three main aims: (1) to describe for the first time the topographic EEG changes of Slow-Wave Activity (SWA) and sleep spindles in a group of children and adolescents with DD, (2) to evaluate the sleep-dependent learning of declarative memories in participants with DD, and (3) to investigate the relation between sleep features and memory performance.

## 2. Materials and Methods

### 2.1. Participants 

Twenty-nine children and adolescents (9–14 years old) were involved in the study: 11 participants with DD (mean age: 11.04 ± 1.67; 5 males) and 18 chronological-age matched NR (mean age: 11.72 ± 1.56; 11 males). Four participants (1 DD and 3 NR) had to be excluded from EEG analyses due to technical difficulties during the polysomnographic (PSG) recording; hence the final sample for EEG analyses included 10 participants with DD and 15 NR.

Participants with DD were recruited at the Child and Adolescents Neuropsychiatric Unit of the Bambino Gesù Children’s Hospital, while NRs were recruited through advertisements in the community.

Before entering the study, all participants were assessed for intelligence, reading abilities, psychiatric and sleep comorbidities by means of a screening test battery.

The absence of intellectual disabilities was assessed through Raven’s Matrices [42] (colored or standard based on age) or Wechsler Intelligence Scale for Children Fourth Edition (WISC-IV; [43]): none of the participants had an IQ score below 85. For reading abilities, the text reading test (MT-2; [44]) and the word and non-word reading tests (DDE-2; [45]) were administered. In order to exclude the presence of psychiatric comorbidities and sleep problems, the Child Behavior Checklist (CBCL) [46] and the Sleep Disturbance Scale for Children [47] were filled out by parents. All participants were free of any medication for the week preceding the experimental night. 

Participants and their parents were informed about the procedure and the aims of the study and gave their written informed consent. At the end of the session, they received a gift. The study has been approved by the Institutional Review Board of the Department of Psychology (#0001266) and was conducted in accordance with the Declaration of Helsinki.

### 2.2. Procedure 

Participants who fulfilled the inclusion/exclusion criteria required for participation in the study underwent a complete PSG recording of a nocturnal sleep, performed at the Sleep Psychophysiology Laboratory of the Department of Psychology (Sapienza, University of Rome).

During the week preceding the experimental night, participants were asked to fill out sleep diaries every morning within 15 min after awakening in order to monitor the regularity of the sleep-wake cycle. They were asked to avoid naps in the day of the experimental night. Each participant took part in the experimental session alone and each on a different day. Participants arrived at the sleep lab between 17:00 and 19:00 and, at first, performed a declarative memory task (encoding and immediate recall phases), that consists of a Word-Pair Task (WPT). After the application of the PSG, participants had a light dinner. Sleep PSG recordings started between 21:00 and 23:00, according to the habitual bedtime of the participants, and finished after 10 h of sleep or with the spontaneous awakening of the subjects. In the morning, the subjects filled out a sleep diary and after 45 min from the awakening the WPT was newly administered (delayed recall phase) (please refer to the Measures Section for more details). Each session lasted about 15 h. Data recordings were conducted on weekends or free-school days. Environmental influences that could disturb sleep (light, noise, temperature) were controlled.

### 2.3. Measures

#### 2.3.1. Declarative Memory Word-Pair Task 

To test the declarative memory consolidation, a WPT modified ad hoc for the present study was used. It consisted of a list of 34 semantically unrelated word pairs that were presented out loud in a randomized order. The auditory version of the task [48,49], instead of the more classical visual one, was chosen and constructed with Psychtoolbox 3 for Matlab R2014a (The Mathworks, Inc., Natick, MA, USA), in order to limit potential encoding difficulties for the participants with DD. The first and the last two word-pairs of the list served to buffer primacy and recency effects and were excluded from the calculation of the performance.

Words were selected from the “Lexvar” Italian database of child language [50]. They had to be bi- or three-syllabic and characterized by high imaginability and concreteness and by a low emotional connotation. To control for mnemonic strategies, participants were instructed to visually imagine a relation between the two randomly related words (e.g., for “cake-river” one could imagine a cake floating on a river) [51]. In the evening, the entire list was presented (encoding session). The time interval between the two words of each pair (intra-stimulus interval) had a 1-s duration, while the interstimulus interval (between one pair and the following) lasted 5 s. The presentation of the list was immediately followed by the cued recall session (immediate recall). In this phase, only the first word of each pair was presented, and subjects were asked to name the second word out loud within 15 s. The experimenter gave feedback to the participants, in terms of right or wrong answer. The list was presented repeatedly, at most four times (or trials), until the subjects remembered at least 50% of the word-pairs (items). The delayed recall session was performed in the morning, 45 min after lights on.

The overnight memory consolidation was calculated as the number of correctly retrieved words at the morning recall referred to the number of correctly retrieved words at the last repetition of evening recall (consolidation rate) [52].

The other WPT parameters considered for evaluating the memory performance were the following:
number of trials (repetition of the list) needed to achieve the cutoff (50% of pairs recalled);number of correctly retrieved word-pairs in the last trial in the evening (immediate recall);number of correctly retrieved word-pairs in the morning (delayed recall).

#### 2.3.2. Polysomnographic Recordings 

The PSG recordings were acquired in a sound-proof, temperature-controlled room, with a Brain Amp MR plus system (Brain Products GmbH, Gilching, Munich, Germany). 

The 28 unipolar EEG derivations of the international 10-10 system (C3, C4, Cp1, Cp2, Cp5, Cp6, Cz, F3, F4, F7, F8, Fc1, Fc2, Fc5, Fc6, Fp1, Fp2, Fz, O1, O2, Oz, P3, P4, P7, P8, Pz, T7, T8) were recorded from scalp electrodes (Ag/AgCl) with averaged mastoid references (A1 and A2). EEG signals were acquired with a sampling rate of 250 Hz and bandpass filtered at 30 Hz. Electrode impedance was kept below 5 kΩ. The bipolar electrooculogram (EOG) electrodes were placed about 1 cm from the lateral canthi up for the right eye and down for the left eye and were recorded with a time constant of 1 s. Submental electromyogram (EMG) was recorded with a time constant of 0.03 s. EEG data were digitally stored for further offline analyses.

### 2.4. Data Analyses 

#### 2.4.1. Demographic and Clinical Characteristics

All data were analyzed using Matlab 2011b (The Mathworks, Inc., Natick, MA, USA). Demographic, cognitive and clinical characteristics (Age, IQ, reading abilities, CBCL, and Sleep Disturbance Scale for Children (SDSC) scores) were compared between the two groups by means of two-tailed *t*-tests for independent samples. Alpha level was set at 0.05. 

#### 2.4.2. Declarative Memory Performance

Regarding memory performance, the encoding variables (number of trials and number of recalled word-pairs in the evening) were compared between the groups to control for potential performance differences at the baseline. The crucial WPT variable (consolidation rate) was calculated as the number of correctly retrieved words at the morning recall as a function of the number of correctly retrieved words at the last repetition of evening recall. It was compared between the two groups by means of two-tailed unpaired *t*-tests. Alpha level was set at 0.05.

#### 2.4.3. Sleep Measures

Sleep stages were visually scored in 20 s epochs, according to Rechtschaffen and Kales’ criteria [53], excluding ocular, muscle, and electrical artifacts. The following macrostructural variables were calculated: stage 1, stage 2 and Slow Wave Sleep (SWS) latency (min); stage 1, stage 2, SWS, REM and NREM (stage 2 + SWS) duration (min); Total Sleep Time (TST, min), given by the sum of stage 1, stage 2, SWS and REM duration; Total Bed Time (TBT); Wake After Sleep Onset (WASO); number of Movement Arousal (MA); Sleep Onset Latency (SOL); Sleep Efficiency Index (SEI) as the ratio between TST and TBT. Differences between the two groups were computed for all the macrostructural variables by means of two-tailed unpaired *t*-tests. Alpha level was set at 0.05.

#### 2.4.4. Slow-Wave Activity (SWA)

For the whole-night NREM sleep, the power spectra for the 28 derivations were computed by Fast Fourier Transform (FFT) for 20 s epochs (periodogram: 4 s), resulting in a frequency resolution of 0.25 Hz. As total spectral power may vary considerably between subjects, EEG power values were normalized for total power and expressed by relative spectral power measures. For each scalp derivation, relative power values for adjacent frequency bins were summed together to obtain the delta frequency band (0.5–4.75 Hz). Differences between groups were computed by means of two-tailed unpaired *t*-tests for the 28 cortical channels. To correct for multiple comparisons the False Discovery Rate (FDR) was applied [54].

#### 2.4.5. Spindle Detection and Analysis

Spindle detection was performed by means of a customized algorithm in Matlab [55,56,57,58,59]. NREM epochs were bandpass-filtered between 11 and 15 Hz (−3 dB at 10 and 16 Hz) using a Chebyshev Type II filter. The detection of a spindle occurred when the mean signal amplitude of each channel exceeded an upper threshold set at six times the mean single channel amplitude. The local amplitude maximum above the upper threshold was considered as the peak amplitude of the single spindle. The points at which the amplitude fell below a lower threshold (two times the mean amplitude of each channel) occurring at least 0.25 s from the peak were considered as the beginning and the end of the spindle (maximum duration: 1.5 s). Spindles falling within the 11–13 Hz frequency range were considered as “slow,” while those falling in the 13–15 Hz range were considered as “fast.” For each participant, the raw number of spindles was calculated for each electrode, separately for fast and slow spindles and jointly for the whole frequency range. Subsequently, spindle density was calculated as the number of spindles divided by artifact-free NREM (stages 2, 3, and 4) sleep minutes.

Differences between groups for spindle density were computed by means of two-tailed unpaired *t*-tests for the 28 cortical channels. To correct for multiple comparisons the FDR was applied.

#### 2.4.6. Correlation between Sleep Measures and Memory Performance

The relations between all the macrostructural variables and the overnight memory consolidation rate were computed by means of Spearman’s correlations. The same analysis was computed for SWA and spindle density for the whole topography (28 cortical derivations) and corrected by means of FDR.

## 3. Results

### 3.1. Demographic and Clinical Characteristics

Means, standard deviations and between-group comparisons (*t*-tests) for demographic and clinical characteristics of the groups are reported in Table 1. The two groups did not differ significantly in age (*t*_1,27_ = −1.13; *p* = 0.269), IQ (*t*_1,27_ = 0.14; *p* = 0.9) or psychiatric comorbidities (internalizing problems: *t*_1,27_ = −0.25; *p* = 0.81; externalizing problems: *t*_1,27_ = −0.07; *p* = 0.95; total problems: *t*_1,27_ = 0.05; *p* = 0.96). As expected, children and adolescents with DD had worse reading abilities than NR in each reading measure considered (MT-2 text reading speed: *t*_1,27_ = −5.9, *p* < 0.001; MT-2 text reading errors, *t*_1,27_ = 5.7, *p* < 0.001; DDE-2 word reading speed: *t*_1,27_ = 5.9, *p* < 0.001; DDE-2 word reading errors: *t*_1,27_ = 3.97, *p* ≤ 0.001; DDE-2 non-word reading speed: *t*_1,27_ = 5.7, *p* < 0.001; DDE-2 non-word reading errors, *t*_1,27_ = 5.72, *p* ≤ 0.001). The two groups were homogeneous regarding the incidence of parent-reported sleep problems (see Table 2). According to the exclusion/inclusion criteria, the two groups were free of psychopathological or sleep problems, assessed through parent-report questionnaires.

### 3.2. Word Pair Task 

Measures of memory performance are listed in Table 3. Participants with DD and NR did not differ in the numbers of trials to criterion nor in the final number of correctly encoded word-pairs in the evening. On the other hand, the group with DD showed a significantly lower sleep-dependent consolidation rate (*t*_1,27_ = −2.501; *p* = 0.019) and a significantly lower number of recalled word pairs in the morning (*t*_1,27_ = −2.125; *p* = 0.043) compared to the NR.

### 3.3. Sleep Macrostructure

Table 4 reports the sleep macrostructural variables in the participants with DD and NR and the results of the between-group statistical comparisons (*t*-tests). No significant differences were found between the two groups for any variable. 

### 3.4. SWA

The topographical distribution of mean SWA during NREM sleep for the two groups is reported in Figure 1A. As expected, SWA is predominantly located in the fronto-central areas in both groups. Statistical comparisons computed by means of unpaired *t*-tests (Figure 1B) failed to reach significance after FDR corrections, although there was a trend in the direction of an increased SWA in the group with DD at the Fc5 (*t*_1,23_ = 1.97 *p* = 0.06) derivation and a decreased SWA at the Cp2 derivation (*t*_1,23_ = −2.76 *p* = 0.01).

### 3.5. Sleep Spindles

Figure 2A shows the topographical distribution of mean sleep spindle density in the two groups and the relative statistical comparisons. The topographical distribution of both fast and slow spindles is consistent with the pre-existing knowledge on this topic [60,61,62] with slow spindles exhibiting their maxima in the frontal areas, while fast spindles peak on more posterior derivations, predominantly central.

After FDR correction, the *t*-tests showed the absence of significant differences between the two groups for spindle density in the whole frequency range and the fast spindles range (Figure 2B). On the other hand, slow spindle density was significantly lower (FDR q ≤ 0.043; *p* ≤ 0.0077 corresponds to a *t* ≥ 2.92 in the group with DD compared with NR at Fc1 (*t*_1,23_ = −3.06; *p* = 0.0056), O1 (*t*_1,23_ = −3.39; *p* = 0.002), Oz (*t*_1,23_ = −2.92; *p* = 0.0077), P7 (*t*_1,23_ = −3.00; *p* = 0.006) e P8 (*t*_1,23_ = −3.14; *p* = 0.0046).

### 3.6. Correlations between Memory Performance and Sleep Measures

#### 3.6.1. Macrostructural Parameters x Memory Performance

No correlations between memory consolidation rate and sleep macrostructural parameters were significant in the group with DD or in NR (Table 5).

#### 3.6.2. SWA and Memory Performance

Figure 3 shows the topographical distributions of correlation coefficients (Rho values) between WPT consolidation rate and SWA of NREM sleep. There were no significant correlations that survived FDR correction, even if some trends can be observed. In particular, in NR there is a positive correlation at F3 (Rho = 0.60; *p* = 0.02), while a negative one is observed at P4 (Rho = −0.70; *p* = 0.003). In the group with DD no correlation was statistically significant.

#### 3.6.3. Sleep Spindle Density and Memory Performance

Figure 4 reports the topographical distributions of the correlation coefficients (Rho values) between spindle (all, fast and slow) density and memory consolidation, separately for the two groups. No significant correlation was found. However, the topographical distribution of the correlational patterns should be noted: in the group with DD negative correlation coefficients are localized mainly in the anterior areas and the positive coefficients in the posterior ones, while in the NR group the correlations have a positive direction diffusely on the scalp.

## 4. Discussion

To the best of our knowledge, the current study is the first to investigate sleep-dependent declarative memory consolidation in children and adolescents with DD considering the topographical aspects of NREM EEG hallmarks. According to our results, participants with DD exhibit a significant worsening in declarative memory performance after a night of sleep compared to age-matched NR, starting from similar baseline levels. Regarding sleep, the differences between the two groups are observable only considering the microstructural features and their topographical distribution, in terms of a net decrease of slow sleep spindles in children and adolescents with DD compared to NR, particularly in the posterior areas. Sleep quality and macrostructural characteristics do not differ between the groups.

### 4.1. Sleep-Dependent Memory Performance

The group with DD shows a reduced consolidation rate at the WPT, compared to NR, starting from a baseline level (number of recalled word-pairs in the evening and number of trials to the criterion) similar between the groups. The absence of differences at the encoding phase seems to confirm the appropriateness of the adjustments applied to the classical WPT (i.e., auditory version). The worsening in the performance appears at the morning recall and we hypothesize that it could be linked to peculiar sleep processes involved in memory consolidation.

Previous studies that investigated long-term declarative memory in DD provided conflicting findings. Some authors showed impaired long-term declarative memory abilities [29,32,33,34] reporting widespread deficits (e.g., [29]), while others failed in showing any difference between individuals with DD and controls [26,30,31].

However, the aforementioned studies did not analyze the potential role of sleep in this process. Only Smith et al. (2017) [20] evaluated sleep-dependent declarative memory consolidation in DD, through the administration of a novel word learning task, suggesting a less active role of sleep in this process in preadolescents with DD compared to NR. Looking at the absence of differences at the baseline performance between participants with DD and NR, our results are partially consistent with that finding. We hypothesize that DD might be characterized by an alteration during the sleep-dependent consolidation processes more than by a declarative memory deficit per se. Nevertheless, the involvement of sleep in the observed deficit cannot be confirmed, as our study lacks a parallel protocol with a wake- instead of a sleep-interval between the immediate and the delayed recall.

### 4.2. Sleep Problems and Macrostructure

The occurrence of sleep problems, parent-reported through the SDSC questionnaire, is similar between the two groups. The only previous study [36] that investigated this aspect showed an increased sleep problem percentage in children with DD compared to controls. Although the involved sample was very large, participants were not assessed for psychopathological comorbidities, very often found in DD [63] and typically associated with sleep problems [64].

Concerning sleep architecture, we have not found any difference between the groups. Earlier findings are heterogeneous. A first study in this field [39] found changes in sleep architecture in DD. However, these previous results are scarcely comparable to the present findings (e.g., the mean IQ in the DD group was lower in the study by Mercier et al. than in our study). More recently, Bruni et al. (2009a; 2009b) [37,38] showed a decrease in SWS and REM sleep percentage, along with an increase of stage 2. On the other hand, the results of Smith et al. (2017) [20] are in line with our findings. In our opinion, the different age range might explain this heterogeneity, since sleep architecture undergoes large modifications during development [9]. While Smith et al. (2017) [20] studied a sample comparable to ours in terms of age (8–13 years), in the studies of Bruni et al. (2009a,b) [37,38] the age range was widespread (8–16).

To summarize, we hypothesize that the sleep disturbances and the alterations of sleep quality and macrostructure previously found in populations with dyslexia might not be part of the phenotype per se but might be due to the presence of psychopathological comorbidities [36,39] and/or to methodological issues [37,38].

### 4.3. Cortical Sleep Topography

The present study is the first that analyzed the topographical distribution (on 28 cortical channels) of sleep spindles and SWA in DD. Our main finding is a significant decrease of bilateral occipito-parietal and left fronto-central slow spindles in the group with DD compared to NR.

A decrease of slow spindles in DD is shown for the first time. Bruni et al. (2009a) [37] reported an increased 11–12 Hz slow EEG activity, which mostly corresponds to the frequency range of slow spindles, and an increased spindle density limited to stage 2 and measured over a single central derivation, both positively related to the degree of reading difficulties. The authors interpreted these patterns in terms of an enhanced role of spindles in the transfer of information between the damaged areas of the posterior left hemisphere and the posterior and anterior vicariant areas, localized in the right hemisphere. Nevertheless, this latter point might appear counterintuitive in light of the knowledge about the role of sleep spindles in the enhancement of cognitive performance [65]. From a topographic point of view, Bruni et al. (2009a) [37] investigated spindle characteristics only on the channel Cz, while in the present work, the main difference between the groups has been found in the posterior areas. Moreover, the authors performed a visual and not an automatic spindle detection, without differentiating between slow and fast spindles. On the other hand, Smith et al. (2017) [20] did not find differences in spindle density between participants with DD and NR. In this case, density was calculated for central and frontal derivations and subsequently averaged across the considered channels, so that no topographical information was provided. Moreover, they did not distinguish between slow and fast spindles and considered a smaller frequency range (12–15 Hz).

To sum up, specific changes of sleep spindle activity in DD might become observable only when considering the topographical distribution and distinguishing between the subtypes, aspects that have been introduced for the first time in our study. This evidence might open the way to a new approach to the study of sleep-related aspects in NDDs, aimed to highlight the local features of EEG activities. This is in line with the local sleep theory [40,41], which claims that several regional EEG phenomena during sleep are expressed proportionally to the brain activity during the previous wake period with a frequency- and topographic-specificity.

The specific functional role of centro-parietal fast and frontal slow spindles is still debated but what is agreed is their positive correlation with cognitive functioning and, specifically, memory consolidation [7,65]. However, a possible distinction in the functional role of the two subtypes during healthy development has yet to be clarified [9]. At a network level, it has been suggested that fast spindles are related to thalamo-cortical couplings, whereas slow spindles are related to cortico-cortical couplings [66]. Starting from this assumption, the reduction in slow spindle density, observed in our study, might reflect the disconnection deficit that has been hypothesized to be at the base of DD [67,68]. A potential cause of this reduced connectivity might be an alteration of white matter. One of the first works in the field [69] showed white matter alterations in subjects with DD compared to NR in bilateral temporo-parietal areas, which are involved in the connection between anterior and posterior regions. Moreover, the integrity of white matter pathways in the left temporo-parietal area has been related to reading abilities (e.g., [70]). Finally, individual differences in sleep spindles might depend at least in part on the structural properties of white matter tracts [71]. According to this view, the posterior reduction of slow spindles in participants with DD observed in our study may be the electrophysiological marker during sleep of the white matter alteration in this population. Future studies on structural and functional connectivity during sleep should directly assess this issue.

Concerning SWA, albeit our findings do not reach statistical significance, they suggest a clear antero-posterior dissociation, showing a trend of increased fronto-central SWA paralleled by a posterior decrease in DD, compared to NR. The anterior delta enhancement is consistent with the central increased SWA observed by Bruni et al. (2009a) [37], extending their finding to several anterior cortical sites. On the other hand, SWA in the posterior regions has never been investigated in DD. Interestingly, individuals with DD compared to NRs show a hypoactivation of two brain reading systems, localized in the parieto-temporal and the occipito-temporal regions of the left hemisphere, and some evidence highlights a concurrent hyperactivation of vicariant systems, in the frontal area bilaterally and the right hemisphere [72,73]. Starting from these findings, the hypothesis that the differences in SWA observed in our study may mirror the functional abnormalities observed in neuroimaging studies represents an intriguing speculation and should be directly investigated.

A limitation of the present study is the absence of an EEG-recorded baseline night without the administration of a cognitive task before sleep. As stated before, the processing of new information that takes place during sleep involves sleep spindles and SWA. The observed differences between the groups in sleep spindles and SWA topographical patterns might be explained by different memory consolidation processes, as suggested by the memory performance data (this point will be further discussed in the next session). This implies that the generalizability of our results on the brain topography should be taken with caution, since some changes may be related to the administration of the declarative task before going to sleep. Future studies should investigate baseline sleep microstructure in this population with a topographical approach.

### 4.4. Relations between Sleep Features and Memory Performance

Macrostructural variables are not significantly related to the WPT performance. This is not surprising as the relation between this kind of sleep parameter and declarative memory consolidation in developmental age is not well-established [9,74].

On the other hand, the correlations between NREM hallmarks and memory performance, albeit not significant, are completely original. Indeed, the correlational patterns are strongly differentiated between the two groups.

Regarding SWA, the coefficients are positive in the left frontal region and negative in the right posterior areas for the NR, while in the group with DD the negative coefficients are located only on the left hemisphere and the positive ones are shifted more centrally and on the left. This might mean that cortical areas involved in sleep-dependent memory consolidation are differentiated between the two groups. The topographical dissociation could be connected to the functional alteration of the left hemisphere networks in DD [72] and/or to the application of unalike mnemonic strategies, that would mirror themselves at a local level during sleep. On the other hand, the stronger correlations in the NR might reflect a more passive role of sleep in the group with DD, consistently with the hypothesis of Smith et al. (2017) [20], who showed significant correlations between memory consolidation and SWA in NR but not in preadolescents with DD.

In the NR the relations between spindle density and memory performance are generally positive, partially consistent with the already existent knowledge about the relation between spindle and memory consolidation, even if in children the evidence is quite heterogeneous [51,62]. On the contrary, the correlational pattern appears dissociated in the group with DD, with positive coefficients located on the posterior areas, concurrently with a negative direction of the effects on frontal areas. Therefore, spindle activity might in this case interfere with memory consolidation processes. Smith et al. (2017) [20] found a moderate correlation (not statistically significant) between power spectra in the spindle frequency range and the performance in the recall phase of a declarative memory task in the control group but not in participants with DD. Albeit our results are scarcely comparable with those of Smith and coworkers (2017) [20] (i.e., different tasks; absence of topographical information), taken together they point to the need for a further effort in the assessment of the relation between cognitive functioning and sleep spindles in DD, particularly considering its regional aspects. In light of this aim future studies might focus on brain oscillation dynamics and connectivity by means of pattern recognition methods capable to modelling several EEG data in parallel, with a concomitant evaluation of local and global dynamics (e.g., [75]).

## 5. Conclusions

Our results represent the first exhaustive description of the sleep EEG topography in DD after declarative learning, showing a worse sleep-dependent memory consolidation and reduced posterior slow spindles activity. Even if in the present study memory performance and sleep EEG were not found to be directly associated, the observed topographical patterns of correlational trends are promising and need further exploration in the near future. However, several methodological choices may have affected our results. The main limitation of the present study is the absence of a baseline night, due to the difficulties in the recruitment of the sample for age and inclusion/exclusion criteria issues. The latter aspect was particularly critical in recruiting participants with DD for the frequent contemporary presence of reading deficits and psychopathological problems. Beyond considering a baseline night, future research should include larger samples of participants to differentiate groups of DD with distinct characteristics.

Our results suggest that only a topographical approach might shed light on potential alteration in cortical oscillation dynamics in NDDs and, consequently, a global view of the sleep phenomenon appears not particularly informative in this field. Functional and structural brain alterations in learning disabilities might be paralleled by localized alteration of sleep microstructural features that become clearly observable only at a local level during sleep. On this basis, sleep EEG oscillations might represent a target for therapeutic interventions aimed at enhancing cognitive (in this case memory) functioning in DD, in the vein of the already consolidated non-invasive brain stimulation interventions [76,77,78,79], potentially effective in improving reading abilities in DD.

## Figures and Tables

**Figure 1 brainsci-11-00073-f001:**
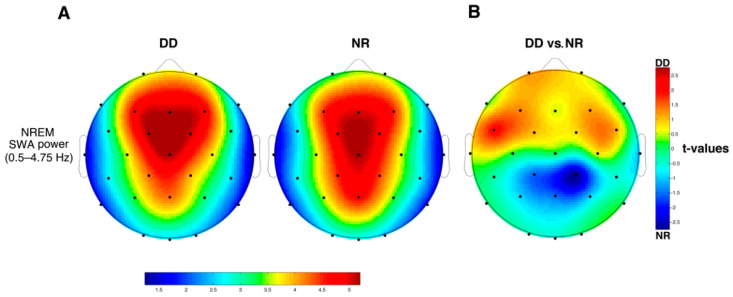
(**A**) Topographical distribution of relative EEG power of delta (0.5–4.75 Hz) activity during NREM sleep for the group with DD and the NR. The average values are color-coded, plotted at the corresponding position on the planar projection of the scalp surface and interpolated between electrodes. The maps are based on 28 EEG derivations of the international 10-10 system with linked mastoid reference, and then normalized to average power across the individual EEG maps. (**B**) Results of the statistical comparisons (two-tailed *t*-tests for independent samples) on EEG delta power of NREM sleep between the group of DD (*n* = 10) and the NR (*n* = 15). The statistical map reports the t-values for each scalp location.

**Figure 2 brainsci-11-00073-f002:**
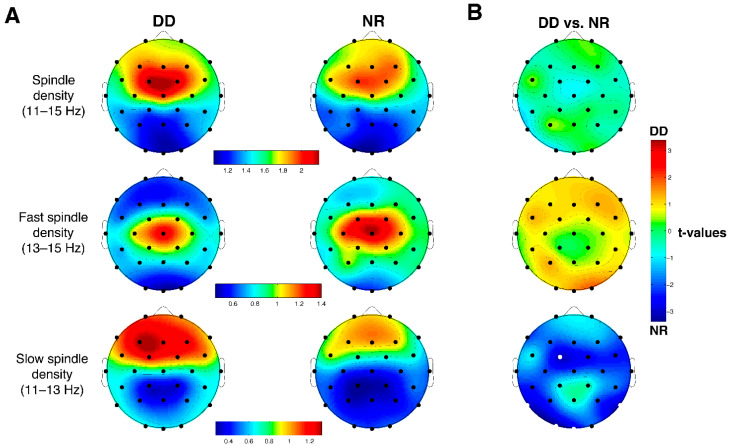
(**A**) Whole frequency range (11–15 Hz), fast (13–15 Hz) and slow (11–13 Hz) spindle density topographical scalp maps in the group of DD and NR. The maps are based on the 28 derivation of the 10-10 system (electrodes positions indicated by black dots). Values are color-coded and plotted at the corresponding position on the planar projection of the hemispheric scalp model. Values between electrodes were interpolated. Values are expressed in terms of number of spindles divided by artifact-free NREM sleep minutes. (**B**) Results of the statistical comparisons (two-tailed *t*-tests for independent samples) on EEG delta power of NREM sleep between the group with DD (*n* = 10) and the NR (*n* = 15). The statistical map reports the t-values for each scalp location. White dots indicate significant differences after FDR correction (*p* < 0.0077).

**Figure 3 brainsci-11-00073-f003:**
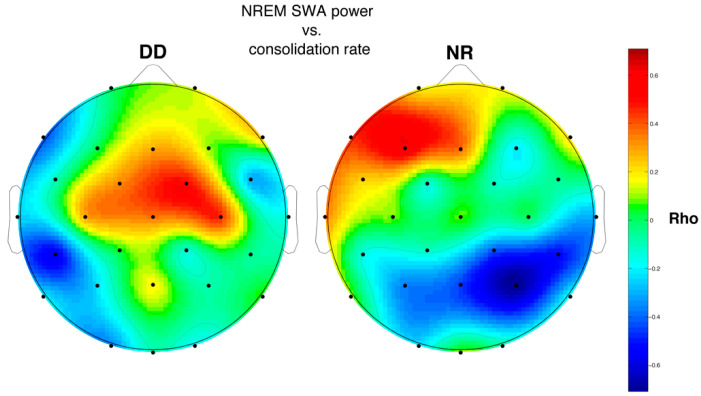
Maps of the correlation coefficients (Spearman’s Rho) between the EEG power of SWA for each electrode (28 channels) and the sleep-dependent consolidation rate at Word pair task (WPT) for the group with DD (*n* = 10) and the NR (*n* = 15). The values of the correlation coefficients are color-coded, plotted at the corresponding position on the planar projection of the scalp surface, and interpolated between electrodes.

**Figure 4 brainsci-11-00073-f004:**
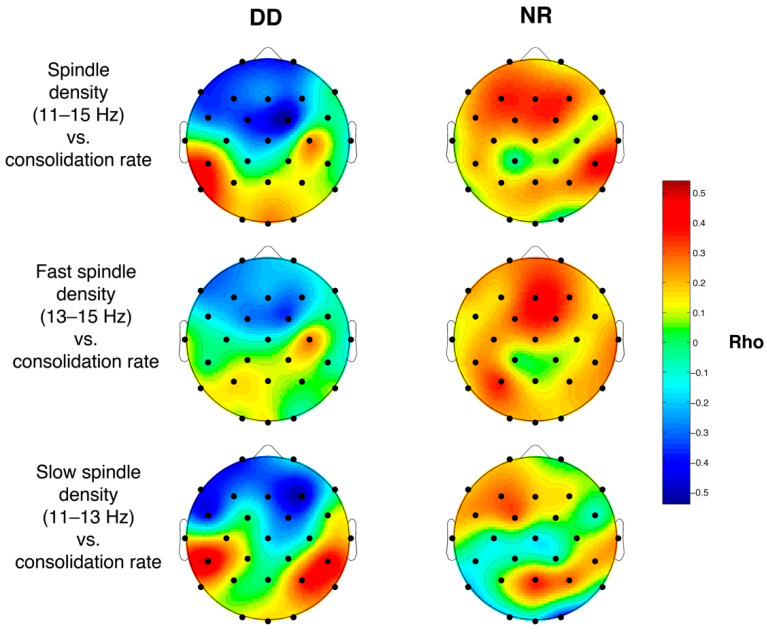
Maps of the correlation coefficients (Spearman’s Rho) between spindle density (all, fast and slow) for each electrode (28 channels) and the sleep-dependent consolidation rate at Word pair task (WPT) for the group with DD (*n* = 10) and the NR (*n* = 15). The values of the correlation coefficients are color-coded, plotted at the corresponding position on the planar projection of the scalp surface, and interpolated between electrodes.

**Table 1 brainsci-11-00073-t001:** Means and standard deviations (SD) of demographic data (age), Intelligence Quotient (IQ), reading (DDE-2 and MT test) and clinical (CBCL) measures for children with Developmental Dyslexia (DD) and Normal Readers (NR). Results of between-group comparisons are also reported (*t* and *p* values). Significant differences between groups (*p* < 0.05) are indicated in bold.

	DD Mean (SD) (*n* = 11)	NR Mean (SD) (*n* = 18)	*t* _(1,27)_	*p*
Age (years, months)	11.04 (1.67)	11.72 (1.56)	−1.13	0.269
IQ	111.3 (6.21)	110.72 (12.37)	0.14	0.9
DDE-2 Word reading				
*Speed* ^1^	181.46 (69.87)	76.74 (23.35)	**5.90**	**<0.001**
*Errors*	8.45 (6.36)	1.72 (2.7)	**3.97**	**<0.001**
DDE-2 Non-word reading				
*Speed* ^1^	119.96 (39.4)	57.7 (19.4)	**5.7**	**<0.001**
*Errors*	14.54 (5.34)	4.44 (4.13)	**5.72**	**<0.001**
MT-2 Text reading				
*Speed* ^2^	1.90 (0.92)	4.66 (1.2)	**−5.9**	**<0.001**
*Errors*	15.10 (5.28)	3.5 (3)	**5.7**	**<0.001**
CBCL				
*Internalizing probl.*	50.6 (10.6)	52.2 (12.4)	−0.25	0.81
*Externalizing probl.*	48.4 (10)	48.8 (10.64)	−0.07	0.95
*Total probl.*	50.6 (9.58)	50.3 (11.5)	0.05	0.96

^1^ seconds; ^2^ syllables/seconds. Abbreviations: DDE-2, Word and Non-word reading tests from the Battery for evaluating Dyslexia and Dysortography-2; MT-2, Text reading test; CBCL, Child Behaviour Checklist.

**Table 2 brainsci-11-00073-t002:** Means and standard deviations (SD) of the Sleep Disturbance Scale for Children (SDSC) parent-report questionnaire, composed by six subscales and a total score, for children with Developmental Dyslexia (DD) (*n* = 11) and Normal Readers (NR) (*n* = 18). Results of between-group comparisons (*t* and *p* values) are also reported.

SDSC	DD Mean (SD) (*n* = 11)	NR Mean (SD) (*n* = 18)	*t* _(1,27)_	*p*
DIMS	11.91 (3.48)	10.67 (2.70)	1.08	0.291
SBD	3.73 (0.79)	4.22 (1.00)	−1.39	0.175
DA	3.36 (0.50)	3.33 (0.49)	0.16	0.873
SWTD	8.73 (2.10)	8.22 (2.21)	0.61	0.548
DOES	8.00 (2.05)	7.44 (3.18)	0.52	0.611
SHY	2.55 (0.93)	2.89 (1.49)	−0.68	0.500
Total Score	38.27 (6.17)	36.78 (7.36)	0.56	0.578

Abbreviations: DIMS, Disorders of initiating and Maintaining Sleep; SBD, Sleep Breathing Disorders; DA, Disorders of Arousal; SWTD, Sleep-Wake Transition Disorders; DOES, Disorders of Excessive Somnolence; SHY, Sleep Hyper-hydrosis.

**Table 3 brainsci-11-00073-t003:** Means and standard deviations (SD) of Word-Pair Task (WPT) variables for participants with Developmental Dyslexia (DD) (*n* = 11) and Normal Readers (NR) (*n* = 18). Results of between-group comparisons, calculated using a two-tailed *t*-test for independent sample, are also reported (*t* and *p* values). Significant differences between groups (*p* < 0.05) are indicated in bold. #: number.

WPT Variables	DD Mean (SD)	NR Mean (SD)	*t* _(1,27)_	*p*
# of trials to criterion	3 (0.89)	2.5 (0.8)	1.578	0.126
# of recalled word pairs in the evening	16.55 (5.88)	19.11 (3.95)	−1.408	0.171
# of recalled word pairs in the morning	14 (6.03)	18.11(4.38)	**−2.125**	**0.043**
Sleep−dependent consolidation rate	0.84(0.14)	0.94 (0.09)	**−2.501**	**0.019**

**Table 4 brainsci-11-00073-t004:** Means and standard deviations (SD) of macrostructural parameters for participants with Developmental Dyslexia (DD) (*n* = 10) and the Normal Readers (NR) (*n* = 15) groups. Results of between-group comparisons, calculated using a two-tailed *t*-test for independent sample, are also reported (*t* and *p* values). Abbreviations: S1, stage 1; S2, stage 2; SWS, Slow Wave Sleep; REM, Rapid Eye Movements; NREM, Non-rapid Eye Movements; WASO, Wake After Sleep Onset; TST, Total Sleep Time; TBT, Total Bedtime; #MA, number of Movement Arousal; SEI, Sleep Efficiency Index; SOL, Sleep Onset Latency.

Macrostructural Variables	DD Mean (SD)	NR Mean (SD)	*t* _(1,23)_	*p*
Latency S1	7.91 (136.80)	13.71 (10.02)	1.66	0.11
Latency S2	11.82 (8.80)	16.63 (11.23)	−1.14	0.27
Latency SWS	25.39 (8.99)	29.34 (14.34)	−0.77	0.45
Latency REM	173.02 (32.21)	193.64 (82.47)	−0.75	0.46
Duration S1	8.00 (5.95)	11.24 (8.64)	−1.03	0.31
Duration S2	280.53 (67.83)	289.78 (58.27)	−036	0.72
Duration SWS	91.43 (25.16)	91.24 (22.97)	0.02	0.98
Duration REM	106.49 (34.33)	104.03 (31.27)	0.19	0.85
Duration NREM	379.96 (56.64)	391.93 (57.08)	−0.52	0.61
% S1	1.70 (1.24)	2.42 (1.93)	−1.04	0.31
% S2	51.17 (8.23)	58.13 (5.73)	−0.34	0.73
% REM	21.56 (4.83)	20.70 (4.69)	0.45	0.66
% NREM	78.44 (4.83)	79.24 (4.78)	−0.41	0.69
% SWS	19.56 (8.08)	18.74 (5.46)	0.30	0.76
WASO	38.80 (31.12)	55.71 (46.31)	−1.01	0.32
TST	486.46 (77.95)	496.24 (75.56)	−0.31	0.76
TBT	534.50 (57.06)	566.35 (46.63)	−1.53	0.14
# MA	53.80 (19.00)	69.93 (27.55)	−1.61	0.12
SEI (%)	90.63 (7.80)	86.58 (8.98)	1.16	0.26
SOL (min)	11.98 (8.86)	16.35 (11.38)	−1.05	0.31

**Table 5 brainsci-11-00073-t005:** Spearman’s correlations between sleep macrostructural parameters and overnight memory recall for each group separately and for the entire sample.

Macrostructural Variables	DD Rho (*p*)	NR Rho (*p*)
Latency S1	0.26 (0.47)	−0.02 (0.95)
Latency S2	0.05 (0.88)	0.03 (0.91)
Latency SWS	−0.05 (0.89)	−0.01 (0.97)
Latency REM	0.03 (0.93)	0.04 (0.89)
Duration S1	−0.33 (0.35)	0.09 (0.74)
Duration S2	−0.36 (0.31)	0.01 (0.98)
Duration SWS	0.24 (0.51)	−0.23 (0.41)
Duration REM	−0.60 (0.07)	−0.18 (0.53)
Duration NREM	−0.26 (0.47)	−0.03 (0.90)
% S1	−0.02 (0.96)	0.11 (0.69)
% S2	−0.08 (0.83)	0.07 (0.81)
% REM	−0.60 (0.07)	−0.19 (0.49)
% NREM	0.60 (0.07)	0.18 (0.51)
% SWS	0.55 (0.10)	−0.07 (0.81)
WASO	0.24 (0.51)	0.16 (0.56)
TST	−0.45 (0.19)	0.07 (0.79)
TBT	−0.50 (0.19)	−0.08 (0.79)
# MA	− 0.33(0.35)	−0.15 (0.58)
SEI (%)	−0.28 (0.43)	−0.14 (0.62)
SOL (min)	0.05 (0.88)	0.05 (0.86)

## Data Availability

The data presented in this study are available on request from the corresponding author.

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
