# Peer review of "Sleep-Related Declarative Memory Consolidation in Children and Adolescents with Developmental Dyslexia"

_brainsci, 2021, doi:10.3390/brainsci11010073_

Round 1

Reviewer 1 Report

The principal objective of the paper is a study to compare sleep-related declarative memory consolidation between development dyslexia (DD) and normal reader subjects. The study approaches the relation between memory performance (a word pair task taken after sleep) and several features estimated from electroencephalographic (EEG) signals acquired during sleep. The paper provides a clear contribution from a practical standpoint. Explanations are comprehensive including discussion on advantages and limitations of the proposed method. Results show topographical patterns for analysis of structural and functional (learning and memory) brain alterations that could be related with sleep alterations. In summary, I consider the contents of the paper are potentially publishable and the following minor issues should be addressed in a revised version of the paper.

Minor changes:

- Please include a reference to the Word-Pair Task (WPT) adapted (line 137).

- The findings of topographical pattern of correlational trends should lead to studies on brain oscillation dynamics and connectivity and their possible automation. Discussion on this subject could be extended, I suggest the following reference: Multichannel dynamic modeling of non-Gaussian mixtures, Pattern Recognition 2019, 93, 312-323.

Author Response

Response to Reviewer 1 comments

The principal objective of the paper is a study to compare sleep-related declarative memory consolidation between development dyslexia (DD) and normal reader subjects. The study approaches the relation between memory performance (a word pair task taken after sleep) and several features estimated from electroencephalographic (EEG) signals acquired during sleep. The paper provides a clear contribution from a practical standpoint. Explanations are comprehensive including discussion on advantages and limitations of the proposed method. Results show topographical patterns for analysis of structural and functional (learning and memory) brain alterations that could be related with sleep alterations. In summary, I consider the contents of the paper are potentially publishable and the following minor issues should be addressed in a revised version of the paper.

We wish to thank the reviewer for his/her description and comments to the study

Point 1: Please include a reference to the Word-Pair Task (WPT) adapted (line 137).

Response 1: Now, we have clarified that our version of the WPT was not adapted from an already existent version but it was created ad hoc for the aims of the present study. The single aspects adopted from other studies are cited (see references 51 and 52) (lines 180 and 191). We added the citations about the auditory form of the task (references 48 and 49, line 172).

Point 2: The findings of topographical pattern of correlational trends should lead to studies on brain oscillation dynamics and connectivity and their possible automation. Discussion on this subject could be extended, I suggest the following reference: Multichannel dynamic modeling of non-Gaussian mixtures, Pattern Recognition 2019, 93, 312-323.

Response 2: We have further discussed this issue and inserted the requested citation (line 590)

Reviewer 2 Report

The current study by Flaminia et al presents interesting and important insights regarding sleep-related declarative memory consolidation in children and adolescents with developmental dyslexia. 

The study is planned and performed well with relevant analysis and conclusion which really advances the field.

There are a few points that need attention-

1- The abstract mentions the aim of the study was to compare sleep-related declarative memory consolidation in 11 participants with DD and 18 normal readers (NR) (9-14 years old) which seems not appropriate. Was the aim specific regarding studying only 11 and 18 participants in each group? This needs revision.

2- As mentioned in the procedure, the sleep analysis was done after performing the word-pair task (WPT) which may or may not affect the normal sleep pattern in the participants. This point needs to be discussed.

3- Were these tests performed in groups or on a single participant at a time? if it was done in groups, were participants allowed to interact with each other before sleep or during dinner time. This is important because if they discussed the test, it will alter their memory and these procedural details will be helpful to add in the methods.

Author Response

Response to Reviewer 2 comments

The current study by Flaminia et al presents interesting and important insights regarding sleep-related declarative memory consolidation in children and adolescents with developmental dyslexia. 

The study is planned and performed well with relevant analysis and conclusion which really advances the field.

We wish to thank also this reviewer for his/her comments to the study.

Point 1: The abstract mentions the aim of the study was to compare sleep-related declarative memory consolidation in 11 participants with DD and 18 normal readers (NR) (9-14 years old) which seems not appropriate. Was the aim specific regarding studying only 11 and 18 participants in each group? This needs revision.

Response 1: Thanks to the reviewer's suggestion, and the sentence was misleading. Accordingly we have rephrased the sentence to make it clearer.

Point 2: As mentioned in the procedure, the sleep analysis was done after performing the word-pair task (WPT) which may or may not affect the normal sleep pattern in the participants. This point needs to be discussed.

Response 2: Again the reviewer is right. Our study had not a sleep recording without WPT. This implies that generalizability of our results on the brain topography should be taken with caution, since some changes may be related to the administration of a declarative task before going to sleep. We have discussed this issue in the revised manuscript (lines 537-545).

Point 3: Were these tests performed in groups or on a single participant at a time? if it was done in groups, were participants allowed to interact with each other before sleep or during dinner time. This is important because if they discussed the test, it will alter their memory and these procedural details will be helpful to add in the methods.

Response 3: Each participant performed the task alone, just before the sleep recording. We have now better detailed this aspect (line 151).